# Verification of the Vitek Reveal System for Direct Antimicrobial Susceptibility Testing in Gram-Negative Positive Blood Cultures

**DOI:** 10.3390/antibiotics13111058

**Published:** 2024-11-07

**Authors:** Giulia Menchinelli, Damiano Squitieri, Carlotta Magrì, Flavio De Maio, Tiziana D’Inzeo, Margherita Cacaci, Giulia De Angelis, Maurizio Sanguinetti, Brunella Posteraro

**Affiliations:** 1Dipartimento di Scienze di Laboratorio ed Ematologiche, Fondazione Policlinico Universitario A. Gemelli IRCCS, 00168 Rome, Italy; giulia.menchinelli@policlinicogemelli.it (G.M.); flavio.demaio@policlinicogemelli.it (F.D.M.); tiziana.dinzeo@unicatt.it (T.D.); giulia.deangelis@unicatt.it (G.D.A.); 2Dipartimento di Scienze Biotecnologiche di Base, Cliniche Intensivologiche e Perioperatorie, Università Cattolica del Sacro Cuore, 00168 Rome, Italy; damiano.squitieri@unicatt.it (D.S.); carlotta.magri@unicatt.it (C.M.); margherita.cacaci@unicatt.it (M.C.); brunella.posteraro@unicatt.it (B.P.); 3Unità Operativa “Medicina di Precisione in Microbiologia Clinica”, Direzione Scientifica, Fondazione Policlinico Universitario A. Gemelli IRCCS, 00168 Rome, Italy

**Keywords:** antimicrobial susceptibility testing, blood cultures, international standard for organization, verification, Vitek Reveal assay

## Abstract

**Background/Objectives**: The International Organization for Standardization (ISO) 20776-2:2021, which replaces ISO 20776-2:2007, focuses solely on the performance of antimicrobial susceptibility testing (AST) assays, emphasizing the ISO 20776-1 broth microdilution method as the reference standard. Consequently, categorical agreement (CA) and associated errors should not be applied. We verified the Vitek Reveal AST assay according to both ISO 20776-2:2021 and ISO 20776-2:2007 criteria. **Methods**: Samples from 100 simulated and clinical Gram-negative (GN) positive blood cultures (PBCs) were tested at a large teaching hospital. The simulated GN-PBCs were obtained from a hospital collection of isolates selected to represent diverse antimicrobial resistance profiles. The Reveal assay results were compared with those from the reference assay, and the time to result (TTR) for the Reveal assay was calculated. **Results**: The essential agreement rates were 96.1% (816/849) for simulated and 98.8% (929/940) for clinical GN-PBC samples. The bias values were −3.1 for simulated and −11.0 for clinical samples. The CA rates were 97.7% (808/827) for simulated and 99.2% (924/931) for clinical samples. The mean TTR ± SD (hours) for resistant organisms was significantly lower (4.40 ± 1.15) than that for susceptible, increased exposure (5.52 ± 0.48) and susceptible (5.54 ± 0.49) organisms. **Conclusions**: Our findings reinforce the potential of the Reveal assay as a valuable tool and support its implementation in clinical microbiology laboratories.

## 1. Introduction

Due to the emergence of new bacterial resistance mechanisms and the introduction of newly approved antibiotics [1], the field of antimicrobial susceptibility testing (AST) is witnessing the development of new tests and systems that are not only more accurate but also capable of delivering faster results than existing methods [2]. Additionally, on 26 May 2022, the European Commission’s In Vitro Diagnostic Regulation (IVDR) introduced a more comprehensive approach to regulating diagnostic devices [3], prompting clinical microbiology laboratories to increasingly focus on verifying the performance of AST assays claimed by manufacturers [4].

The ISO 20776-2:2021 document, issued by the International Organization for Standardization (ISO), specifies the acceptable performance criteria for AST devices used to determine the minimum inhibitory concentration (MIC) values of antimicrobial agents against bacterial isolates in clinical microbiology laboratories [5]. This document, which replaces ISO 20776-2:2007, emphasizes the use of the broth microdilution (BMD) method (ISO 20776-1) [6] as the reference standard and focuses solely on assay performance, without addressing result interpretation. For this reason, categorical agreement (CA) and its associated terminology (e.g., very major error [VME]), as described in the Clinical and Laboratory Standards Institute (CLSI) M52 document [7], are not applied [5].

According to ISO acceptance criteria, we assessed the accuracy of a new rapid AST assay [8], the Vitek Reveal (bioMérieux, Marcy l’Étoile, France) system, which detects the growth of Gram-negative (GN) bacteria via their emission of volatile organic compounds (VOCs) directly from positive blood culture (PBC) samples. The results from the Vitek Reveal AST assay and the ISO 20776-1 BMD method (i.e., the reference AST assay) were compared for both simulated and clinical sets of GN-PBC samples, the first set including bacterial isolates with highly diversified antimicrobial resistance profiles. Any discordant results between the two assays were resolved.

## 2. Results

We evaluated the Vitek Reveal assay results for 849 and 940 antimicrobial agents and GN bacterial organisms in simulated and clinical PBC samples, respectively. Of the included antibiotics, 21 out of 21 were tested against *Escherichia coli* (33 organisms), 19 out of 21 against *Klebsiella pneumoniae*/*Klebsiella oxytoca* (34/1 organisms), 18 out of 21 against *Enterobacter cloacae* (1 organism), 17 out of 21 against *Klebsiella aerogenes*/*Proteus mirabilis* (2/5 organisms), 14 out of 21 against *Pseudomonas aeruginosa* (17 organisms) and 8 out of 21 against *Acinetobacter baumannii* (7 organisms). This resulted in a total of 133 instances in which a single antibiotic was tested against one or more different bacterial species.

As shown in Table 1 and Table 2, the essential agreement (EA) rates by organism (*Enterobacterales*, *P. aeruginosa* or *A. baumannii*) for the simulated GN-PBC samples were 95.5%, 97.4% and 100%, respectively, while the rates for the clinical GN-PBC samples were 98.7%, 100% and 100%, respectively. Looking at the individual antibiotics, cefepime demonstrated the lowest EA rates when tested against *Enterobacterales* organisms in both simulated (eight non-EA results; six with *K. pneumoniae* and two with *E. coli*) and clinical GN-PBC samples (three non-EA results; two with *E. coli* and one with *P. mirabilis*). The EA rates for all organisms were 96.1% and 98.8% for the simulated and clinical GN-PBC samples, respectively. The overall bias values were −3.1 (calculated for 156 organisms with on-scale MICs) for simulated GN-PBC samples and −11.0 (calculated for 86 organisms with on-scale MICs) for clinical GN-PBC samples.

Among the GN bacterial organisms from simulated PBCs (Appendix A), all were resistant to at least one tested antimicrobial, and 48 were classified as multidrug resistant, according to the Magiorakos et al.’s definitions [9]. Twenty-nine organisms were resistant to carbapenems, and ten organisms were ESBL positive. Out a total of four hundred and ninety-one resistant antimicrobial/organism results, six results from the Vitek Reveal AST assay were classified as VME, including aztreonam/*K. pneumoniae* (one result), ertapenem/*E. coli* (one result), gentamicin/*P. mirabilis* (one result), piperacillin/tazobactam/*E. coli* (one result) and piperacillin/tazobactam/*P. aeruginosa* (two results). Among the GN bacterial organisms from clinical PBCs, 42 were resistant to at least one tested antimicrobial, and 26 were classified as multidrug resistant, according to the Magiorakos et al.’s definitions [9]. Six organisms were resistant to carbapenems, and eight organisms were ESBL positive. Out of a total of two hundred and twenty-one resistant antimicrobial/organism results, one result from the Vitek Reveal AST assay was classified as a VME and was associated with tobramycin/*K. pneumoniae*. Excluding organisms with MICs near the 2024 EUCAST breakpoints allowed us to observe that CA rates increased from 97.7% to 99.1% for simulated GN-PBC samples and from 99.2% to 100% for clinical GN-PBC samples. Overall, five results remained as VME, and two (aztreonam/*K. pneumoniae* and piperacillin/tazobactam/*P. aeruginosa*) were resolved.

We analyzed the time to result (TTR) for the Vitek Reveal AST assay for all 100 GN-PBC samples included in the study, stratified according to the susceptible (S)/susceptible, increased exposure (I)/resistant (R) categories of organisms (Figure 1).

The mean TTR ± standard deviation (SD) (hours) for R organisms was significantly lower (4.40 ± 1.15) than that for I (5.52 ± 0.48) or S (5.54 ± 0.49) organisms (*p* < 0.0001 for both comparisons). The differences remained substantial when assessed for organism groups (*Enterobacterales* or *P. aeruginosa*), but statistical significance was retained only for the *Enterobacterales* group (R, 4.32 ± 1.16 versus I, 5.50 ± 0.47; R, 4.32 ± 1.16 versus S, 5.55 ± 0.49; *p* < 0.0001 for both comparisons). For the *Enterobacterales* group, the mean TTR ± SD values of cefepime, cefotaxime, ceftazidime/avibactam, ceftolozane/tazobactam and piperacillin/tazobactam were statistically lower in R organisms than in S organisms (Appendix A). Only for clinical PBC samples, when the mean TTR ± SD (hours) for the Vitek Reveal AST assay (6.34 ± 0.26) was added to the mean time to positivity ± SD (8.52 ± 2.45) and the mean time to identification ± SD (1.09 ± 0.00), the overall mean time to report ± SD for all GN-PBC samples was 16.36 ± 2.50 (Appendix A).

## 3. Discussion

Our study demonstrated that the Vitek Reveal AST assay provides rapid and reliable AST results for GN organisms in PBCs, with EA rates reaching 96.1% for simulated PBC samples and 98.8% for clinical PBC samples. Additionally, the CA rates were excellent at 97.7% for simulated and 99.2% for the clinical samples. Notably, we found that the TTR for R organisms was significantly one hour shorter than that for the I and S organisms, indicating the assay’s potential to support faster therapeutic decision-making in clinical practice.

The novelty of our study lies in its rigorous evaluation of the Vitek Reveal AST assay in accordance with ISO standards [5,6], utilizing the ISO 20776-1 BMD method as a comparator. Unlike previous studies that compared the Vitek Reveal AST assay with various commercial BMD-based AST methods, such as the bioMérieux Vitek 2 AST cards, the Thermo Fisher Sensititre plates or the Merlin Diagnostika Micronaut plates [10,11,12,13,14], our research focuses specifically on the verification process following the ISO 20776-2:2021 guidelines [5]. Compliance with a BMD prepared according to ISO 20776-1 is mandatory whenever a new commercial AST assay is verified [4]. In alignment with the ISO 20776-2:2021 guidelines, we emphasized EA and bias as critical metrics for assessing the assay’s performance, consistent with the latest recommendations in the field [4].

Incorporating direct inoculation from PBCs into AST assays has been explored in previous studies. For instance, in 2004, Bruins et al. [15] investigated the susceptibility testing of *Enterobacteriaceae* and *P. aeruginosa* through direct inoculation from PBC bottles into the bioMérieux Vitek 2 AST cards. Compared with the standard method (i.e., inoculation of Vitek 2 with bacterial isolates subcultured in agar media), the overall MIC agreement among the 312 isolates tested was 99.2%. Their findings highlighted the potential for rapid AST results using this approach. Subsequently, in 2019, our research [16] further evaluated the Vitek 2 system for rapid susceptibility testing of GN bacterial organisms from PBCs. We found that the overall CA between the Vitek 2 and BMD AST assays among the 505 isolates tested was 91.7%, providing results comparable to those obtained from subcultures. However, the Vitek Reveal AST assay offers several advantages over conventional methods. It provides a more reproducible and standardized process, aligning closely with the ISO 20776-1 BMD method, which is subculture-based. This alignment enhances the reliability of AST results and supports more accurate clinical decision-making.

To make our findings comparable with those from previous studies, we reported both CA and associated errors as additional metrics for evaluating our Vitek Reveal assay results. Our study included both simulated and clinical PBC samples, enriching our dataset with results for highly resistant GN bacterial organisms. As expected, both the EA and CA rates in our study aligned closely with those published elsewhere [10,11,12,13,14]. For example, Tibbetts et al. [10] reported EA/CA rates of 97.7%/95.2% and 98.0%/96.3% for their sets of simulated and clinical GN-PBCs, respectively, while Bianco et al. [12] evaluated 2220 antibiotic/organism combinations (48.7% of which were resistant) to show overall EA/CA rates of 97.7% and 97.6%, respectively. In our study, as in others, the antimicrobial resistance profiles observed in simulated samples mirrored those found in clinical isolates. However, it is noteworthy that the simulated samples primarily consisted of multidrug-resistant organisms (48/50 versus 26/50). This indicates that, while the assay performs well for both simulated and clinical samples, clinicians should be cautious in interpreting results from routine clinical samples, as these may not reflect the higher prevalence of resistance observed in the simulated samples.

Advancements in the AST field are crucial, given that delays in initiating appropriate antimicrobial therapy can significantly impact patient outcomes, particularly in cases of bloodstream infections [17]. The study by Yuceel-Timur et al. [18] emphasizes the importance of rapid AST results, as timely intervention is vital in managing such infections. Furthermore, the study by Yusuf et al. [4] discusses how effective verification processes enhance the overall reliability of new AST systems, further supporting our findings. In this context, it is important to note that we minimized variations in conditions between the Vitek Reveal AST assay and the reference AST assay to ensure the accuracy of our evaluation. The provision of accurate MIC results, along with a significantly shorter TTR for antimicrobial-resistant organisms compared to antimicrobial-susceptible organisms, is crucial for timely directing therapeutic choices towards potentially effective antimicrobials.

Despite these promising results, our study has limitations that should be acknowledged. The simulated PBC samples included organisms with highly different resistance profiles, which may not fully represent the complexity of clinical infections. For instance, we did not assess the presence of inducible AmpC enzymes in our samples, nor could we use polymicrobial BC samples. Additionally, the sample size for certain bacterial species (e.g. *E. cloacae*) was limited, potentially affecting the generalizability of our findings. Future research should aim to include a broader range of clinical PBC samples and assess the Vitek Reveal AST assay in diverse healthcare settings to further validate its efficacy. Furthermore, the observed lower EA for cefepime may be attributed to variations in its source, as the USP (Rockville, MD, USA)-supplied cefepime exhibited different performance compared to other antibiotics sourced from Sigma (St. Louis, MD, USA); we did not observe similar issues with other antibiotics, but the unique characteristics of cefepime, including its stability and interaction with bacterial species, could make it more susceptible to such variations. Finally, while our focus on EA and bias aligns with current studies advocating for these metrics over CA [4], the clinical relevance of CA should not be discounted. Clinicians often rely on CA to guide treatment decisions [19], and maintaining a balance between these metrics is essential for effective antimicrobial stewardship.

## 4. Methods

### 4.1. Study Design and Samples

This study was conducted at a large tertiary-care teaching hospital in Rome, Italy, over a 4-month period (April to August 2024), following the protocol outlined in Figure 2. We included simulated (*n* = 50) and clinical (*n* = 50) PBC samples that were obtained after blood culture (BC) bottles (bioMérieux BacT/Alert FA [aerobic] Plus and FN [anaerobic] Plus) were incubated and subsequently flagged as positive by the BacT/Alert Virtuo BC automated system (bioMérieux). Following the detection of microbial growth, Gram staining was performed to confirm the presence of GN bacteria and to assess monomicrobial growth. For clinical PBCs (described below), aerobic or anaerobic bottles (depending on which type flagged as positive first) underwent direct analysis using the bioMérieux BioFire Blood Culture Identification 2 (BCID2) panel assay. This assay was utilized for bacterial species identification and for detecting various resistance genes, including those encoding carbapenemases (*bla*_IMP_, *bla*_KPC_, *bla*_OXA-48_-like, *bla*_NDM_, *bla*_VIM_), extended-spectrum β-lactamases (ESBL; *bla*_CTX-M_), and the gene conferring resistance to colistin (*mcr-1*) [20].

The first set of samples comprised simulated PBCs, generated by inoculating BC bottles with GN organisms, including 33 *Enterobacterales*, 11 *P. aeruginosa* and 6 *A. baumannii* complex. These organisms were selected from a hospital collection of clinical isolates to represent diverse antimicrobial resistance profiles, with one isolate per bottle. As detailed in Appendix A, β-lactam resistance genes other than those above mentioned or gene types (e.g., *bla*_KPC-3_) in these isolates were identified using a well-established PCR-sequencing approach [21]. The spiking procedure for BCs with whole blood and bacterial cells adhered to methods outlined in our previous simulation studies [22]. The second set included clinical PBCs, which were randomly obtained from hospitalized patients during the study period and yielded GN organisms (43 *Enterobacterales*, 6 *P. aeruginosa* and 1 *A. baumannii* complex) with either antimicrobial-susceptible or -resistant profiles. These PBCs represent 25.1% (50/199) of routinely processed GN-PBCs with monomicrobial growth and potentially eligible for the study. Aliquots from each PBC bottle were either directly used for the Vitek Reveal AST assay (within 16 h of bottle positivity, as recommended by the manufacturer) or simultaneously plated on MacConkey (bioMérieux) and 5% sheep blood tryptic soy (bioMérieux) agar media (Figure 2). Overnight-grown isolates from these plates were subsequently used for the BMD reference AST assay.

### 4.2. Vitek Reveal AST Assay

Study samples were diluted 1000-fold in Pluronic water (Beckman Coulter, Brea, CA, USA), and 115 μL volumes of each dilution were inoculated into the wells of Vitek Reveal GN01-AST dried antibiotic plates (bioMérieux). The antibiotics present in the Vitek Reveal GN01-AST plates, as detailed in Appendix A, included amikacin, amoxicillin/clavulanic acid, ampicillin, aztreonam, cefepime, cefotaxime, ceftazidime, ceftazidime/avibactam, ceftolozane/tazobactam, ciprofloxacin, ertapenem, gentamicin, imipenem, levofloxacin, meropenem, meropenem/vaborbactam, piperacillin, piperacillin/tazobactam, tobramycin, tigecycline and trimethoprim/sulfamethoxazole. Each plate was covered with the Vitek Reveal sensor panel (bioMérieux), which consists of a colorimetric seven-sensor array positioned over each well, and sealed with the Vitek Reveal plate sealer (bioMérieux). The sealed plates were labelled with sample barcodes and loaded into the Vitek Reveal instrument (bioMérieux). As microbial growth produces VOCs, the resulting colorimetric changes in the sensor response allow for the determination of MIC. Pathogen identification was performed through direct analysis or on subcultured bacterial colonies from PBC samples using MALDI-TOF mass spectrometry (Bruker Daltonics, Bremen, Germany). The direct analysis identification was then entered into the user interface, enabling the generation of MIC values for the Vitek Reveal AST assay. Reproducibility, defined as the extent to which consistent MIC results were obtained when the Vitek Reveal AST assay was repeated, was assessed using quality control strains, including *E. coli* ATCC 25922, *K. pneumoniae* ATCC 700603, *K. pneumoniae* ATCC BAA-2814 and *P. aeruginosa* ATCC 27853. The ATCC strains were purchased from KwikStik Microbiologics (Saint Cloud, MN, USA). More than 99% of MIC determinations fell within acceptable ranges [5].

### 4.3. Reference AST Assay

Reference minimum inhibitory concentrations (MICs) were obtained by testing the bacterial isolates from the study samples against all the antibiotics used in the Vitek Reveal AST assay at their respective concentrations (Appendix A). This was performed following antibiotic microdilution procedures and using a cation-adjusted Mueller Hinton (caMH) broth (Thermo Fisher, Waltham, MA, USA), as outlined in the ISO 20776-1:2019 version [6]. Briefly, three to five colonies from each isolate were suspended in sterile saline solution (bioMérieux) to achieve a 0.5 McFarland turbidity. The resulting suspension was then diluted in caMH broth to obtain a bacterial cell concentration equivalent to 5 × 10^5^ colony-forming units per millilitre. This suspension was used to inoculate the antibiotic-containing wells (50 µL per well) of a pre-prepared BMD plate. All BMD plates (one for each isolate) were incubated at 35 °C in a 5% CO_2_-enriched atmosphere and subsequently read visually to determine the MIC values for all antibiotics included in this study.

### 4.4. Data Analysis

According to both ISO 20776-2:2007 and ISO 20776-2:2021 documents, the MIC for a given antibiotic (tested alone or in combination) was defined as the lowest concentration that prevented visible bacterial growth within the assay’s time frame. MIC values were generated and interpreted according to the EUCAST breakpoints v14.0 [23]. A difference in a single result between the Vitek Reveal AST assay and the reference AST assay that fell outside the region of essential agreement (EA) or had a different categorical interpretation (as detailed below) was classified as an error. As evident from the ranges of antibiotic concentrations for both assays (Appendix A), the range of reference MIC values was wider than the MIC range of the Vitek Reveal AST assay. To calculate EA, values lower than the lowest MIC or higher than the highest MIC reported by the Vitek Reveal AST assay (i.e., off-scale MICs) were adjusted by combining them with MICs that correspond to the lower or higher ends of the Vitek Reveal AST assay, respectively [5].

The performance of the Vitek Reveal AST assay was evaluated using two key metrics, EA and bias, in accordance with the ISO 20776-2:2021 criteria. The EA measures the alignment between the test method (the Vitek Reveal AST assay) and the reference method (the reference AST assay), defined as MIC results obtained within ± one 2-fold dilution of the reference MIC. An overall EA rate of ≥90% is considered acceptable. To assess bias, isolates were categorized based on their reference MICs. The analysis calculated the percentage of test results exceeding the reference values and those falling below them. The difference between these two percentages serves as the estimate of test bias, with an acceptable range defined as −30% to +30%. A total of 25 isolates with on-scale MICs are required for this analysis, ensuring reliable detection of bias without falsely indicating unacceptable levels. Consistent with this criterion, bias was calculated for antibiotics with at least 10 on-scale MIC results and/or for total antibiotics only in two of the three organism groups included in this study (*Enterobacterales* and *P. aeruginosa*).

The performance of the Vitek Reveal AST assay was also evaluated by calculating CA, defined as agreement on the same categorical interpretation (S, I or R) based on MIC values, in accordance with the 2024 EUCAST breakpoints [23]. If the S/I/R categorization from the Vitek Reveal AST assay did not exactly match that of the reference AST assay, categorical errors were classified as very major errors (VME; falsely susceptible), major errors (ME; falsely resistant), and minor error (mE; increased exposure susceptibility versus susceptible or resistant), according to the ISO 20776-2:2007 criteria.

The TTR for the Reveal Vitek assay was measured in hours and calculated for each antibiotic/organism combination by summing the hands-on time and the time to assay result. Results were stratified according to the S/I/R categories for all organisms included in this study. Statistically significant differences (ANOVA or Student *t*-test, as appropriate; *p* < 0.05) in the TTR (expressed as mean ± SD) were assessed for all antibiotics tested, as well as for the most clinically relevant antibiotics in a multidrug resistance setting. The time to a final report for clinical PBC samples, starting from their collection to the release of AST results, was also calculated.

## 5. Conclusions

The Vitek Reveal AST assay demonstrates a strong performance in accurately assessing AST in GN bacterial organisms, reinforcing its potential as a valuable tool in clinical microbiology. Our findings support its implementation in clinical microbiology laboratories, particularly in settings where timely information is crucial for patient management. Continued exploration of this assay’s performance will be essential for its integration into routine clinical practice. Future studies should further investigate the assay’s applicability across a wider range of organisms and antimicrobial resistance mechanisms, particularly in real-world settings facing significant levels of resistance. The integration of rapid AST systems into routine clinical practice promises to enhance timely and appropriate antibiotic therapy, ultimately improving patient outcomes in the fight against resistant pathogens.

## Figures and Tables

**Figure 1 antibiotics-13-01058-f001:**
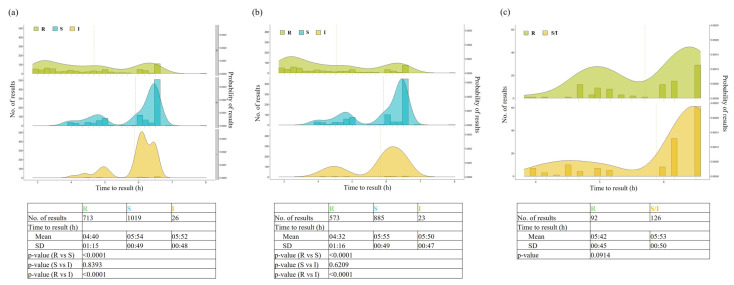
Times to MIC results by Vitek Reveal AST assay for all organisms (**a**), *Enterobacterales* only (**b**) and *Pseudomonas aeruginosa* only (**c**) from positive blood culture samples. Times (range: 3 to 8 h) were stratified according to the S/I/R groups of organisms and presented using histograms with Kernel density plots. Differences between groups (in terms of means ± standard deviation) were assessed for statistical significance. The probability of MIC results falling in the indicated range was calculated by fixing the total area under the curve of each density plot equal to 1.

**Figure 2 antibiotics-13-01058-f002:**
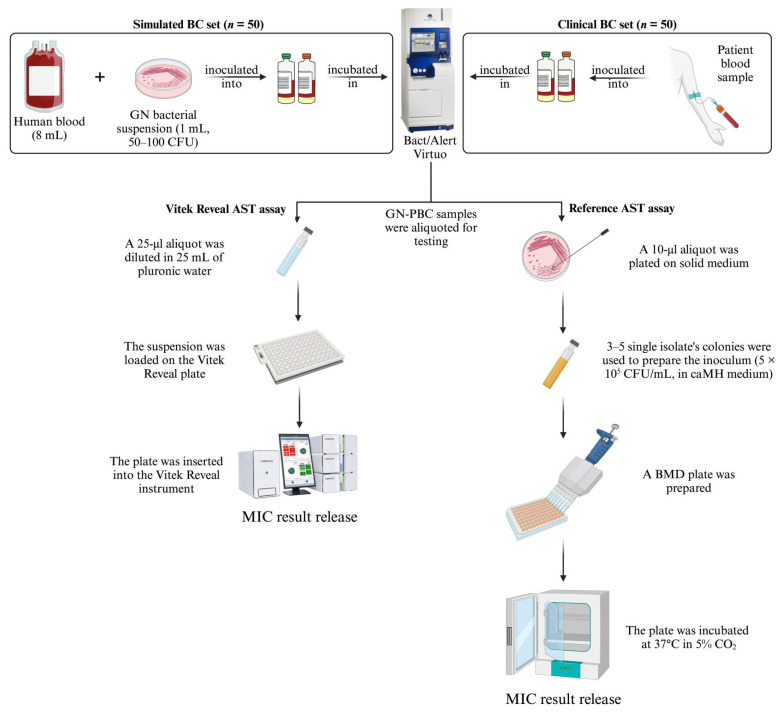
Study protocol description. Two sets of GN-PBC samples were studied. For each set, aliquots from the same sample were simultaneously processed using either the Vitek Reveal AST assay according to the manufacturer’s recommendations or a reference AST assay following the ISO 20776-1:2019 procedure. MIC results were released automatically for one assay and manually for the other assay. AST, antimicrobial susceptibility testing; BC, blood culture; BMD, broth microdilution; caMH, cation-adjusted Mueller Hinton; CFU, colony-forming unit; MIC, minimum inhibitory concentration.

**Table 1 antibiotics-13-01058-t001:** Evaluation of the Vitek Reveal AST assay results obtained with simulated positive blood culture samples for Gram-negative bacterial organisms.

	No. of Organisms Categorized According to EUCAST Breakpoints v14.0 as:			Parameters Evaluated According to:					
				Both ISO 20776–2:2021 and ISO 20776–2:2007	ISO 20776–2:2021 Only	ISO 20776–2:2007 Only			
	R	I	S	EA (%; *n*/*N*)	Bias (%; No. of On-Scale Organisms)	CA (%; *n*/*N*)	ME (%; *n*/*N*)	VME (%; *n*/*N*)	mE (%; *n*/*N*)
*Enterobacterales* organisms (*n* = 33) tested against:									
Amikacin	11	–	22	100; 33/33	–	100; 33/33	0.0; 0/22	0.0; 0/11	–
Amoxicillin/clavulanic acid	28	–	3	100; 31/31	–	100; 31/31	0.0; 0/3	0.0; 0/28	–
Ampicillin	14	–	0	100; 14/14	–	100; 14/14	0.0; 0/0	0.0; 0/14	–
Aztreonam	21	4	7	87.5; 28/32	–	90.6; 29/32	0.0; 0/7	4.8; 1/21	6.2; 2/32
Cefepime	19	5	7	74.2; 23/31	+12.6; 18	96.8; 30/31	0.0; 0/7	0.0; 0/19	3.3; 1/31
Cefotaxime	26	2	5	97.0; 32/33	–	90.9; 30/33	0.0; 0/5	0.0; 0/26	9.1; 3/33
Ceftazidime	26	4	3	90.9; 30/33	+21.4; 12	100; 33/33	0.0; 0/3	0.0; 0/26	0.0; 0/33
Ceftazidime/avibactam	8	–	25	97.0; 32/33	–	100; 33/33	0.0; 0/25	0.0; 0/8	–
Ceftolozane/tazobactam	20	–	12	100; 32/32	–	100; 32/32	0.0; 0/12	0.0; 0/20	–
Ciprofloxacin	26	–	7	100; 33/33	–	100; 29/29	0.0; 0/5	0.0; 0/24	–
Ertapenem	15	–	18	90.9; 30/33	–	97.0; 32/33	0.0; 0/18	6.7; 1/15	–
Gentamicin	13	–	20	100; 33/33	–	97.0; 32/33	0.0; 0/20	7.7; 1/13	–
Imipenem	10	0	22	96.9; 31/32	–	100; 32/32	0.0; 0/22	0.0; 0/10	0.0; 0/32
Levofloxacin	22	1	10	100; 33/33	–	100; 33/33	0.0; 0/10	0.0; 0/22	0.0; 0/33
Meropenem	8	4	21	90.9; 30/33	–	93.9; 31/33	0.0; 0/21	0.0; 0/8	6.1; 2/33
Meropenem/vaborbactam	6	–	27	100; 33/33	–	100; 33/33	0.0; 0/27	0.0; 0/6	–
Piperacillin	32	–	1	100; 33/33	–	100; 33/33	0.0; 0/1	0.0; 0/32	–
Piperacillin/tazobactam	23	–	10	93.9; 31/33	–	96.2; 25/26	0.0; 0/10	6.2; 1/16	–
Tigecycline	0	–	13	100; 13/13	–	100; 13/13	0.0; 0/13	0.0; 0/0	–
Tobramycin	21	–	12	97.0; 32/33	–	97.0; 32/33	8.3; 1/12	0.0; 0/21	–
Trimethoprim/sulfamethoxazole	21	1	11	93.9; 31/33	–	90.9; 30/33	0.0; 0/11	0.0; 0/21	9.1; 3/33
Total antibiotics	370	21	256	95.5; 618/647	−4.3; 108	97.5; 620/636	0.4; 1/254	1.1; 4/361	4.2; 11/260
*Pseudomonas aeruginosa* organisms (*n* = 11) tested against:									
Amikacin	3	–	8	100; 11/11	–	100; 11/11	0.0; 0/8	0.0; 0/3	–
Aztreonam	2	9	–	100; 11/11	–	100; 11/11	0.0; 0/9	0.0; 0/2	–
Cefepime	4	7	–	100; 11/11	–	100; 11/11	0.0; 0/7	0.0; 0/4	–
Ceftazidime	7	4	–	100; 11/11	–	100; 11/11	0.0; 0/4	0.0; 0/7	–
Ceftazidime/avibactam	6	–	5	100; 11/11	–	100; 11/11	0.0; 0/5	0.0; 0/6	–
Ceftolozane/tazobactam	6	–	5	100; 11/11	–	100; 11/11	0.0; 0/5	0.0; 0/6	–
Ciprofloxacin	10	1	–	100; 11/11	–	100; 11/11	0.0; 0/1	0.0; 0/10	–
Imipenem	11	0	–	90.9; 10/11	–	100; 11/11	0.0; 0/0	0.0; 0/11	–
Levofloxacin	9	2	–	100; 11/11	–	–	–	–	–
Meropenem	8	3	0	90.9; 10/11	–	90.9; 10/11	0.0; 0/0	0.0; 0/8	9.1; 1/11
Meropenem/vaborbactam	8	–	3	90.9; 10/11	–	100; 11/11	0.0; 0/3	0.0; 0/8	–
Piperacillin	11	0	–	100; 11/11	–	100; 11/11	0.0; 0/0	0.0; 0/11	–
Piperacillin/tazobactam	8	3	–	90.9; 10/11	–	81.8; 9/11	0.0; 0/3	25.0; 2/8	–
Tobramycin	6	–	5	100; 11/11	–	100; 11/11	0.0; 0/5	0.0; 0/6	–
Total antibiotics	99	29	26	97.4; 150/154	−2.0; 48	97.9; 140/143	0.0; 0/50	2.2; 2/90	9.1; 1/11
*Acinetobacter baumannii* organisms (*n* = 6) tested against:									
Amikacin	4	–	2	100; 6/6	–	100; 6/6	0.0; 0/2	0.0; 0/4	–
Ciprofloxacin	6	0	–	100; 6/6	–	100; 6/6	–	0.0; 0/6	–
Gentamicin	3	–	3	100; 6/6	–	100; 6/6	0.0; 0/3	0.0; 0/3	–
Imipenem	6	0	0	100; 6/6	–	100; 6/6	0.0; 0/0	0.0; 0/6	0.0; 0/6
Levofloxacin	6	0	0	100; 6/6	–	100; 6/6	0.0; 0/0	0.0; 0/6	0.0; 0/6
Meropenem	6	0	0	100; 6/6	–	100; 6/6	0.0; 0/0	0.0; 0/6	0.0; 0/6
Tobramycin	4	–	2	100; 6/6	–	100 6/6	0.0; 0/2	0.0; 0/4	–
Trimethoprim/sulfamethoxazole	5	0	1	100; 6/6	–	100 6/6	0.0; 0/1	0.0; 0/5	0.0; 0/6
Total antibiotics	40	0	8	100; 48/48	–	100; 48/48	0.0; 0/8	0.0; 0/40	0.0; 0/24
All organisms tested (*n* = 50)	509	50	290	96.1; 816/849	−3.1, 156	97.7; 808/827	0.3; 1/312	1.2; 6/491	4.1; 12/295

AST, antimicrobial susceptibility testing; CA, categorical agreement; EA, essential agreement; EUCAST, European Committee on Antimicrobial Susceptibility Testing; I, susceptible, increased exposure; ISO, International Organization for Standardization; ME, major error; mE, minor error; R, resistant; S, susceptible, standard dosing regimen; VME, very major error. Note: The symbol “–” indicates not applicable. Across the tested organism groups, the number of evaluated antimicrobials varies by bacterial species, as species known to exhibit intrinsic resistance to the antimicrobials included in the Vitek Reveal AST assay, or for which EUCAST breakpoints are unavailable, were excluded from the analysis. The EUCAST-defined category “area of technical uncertainty” only applies to two of the antibiotics listed here (ciprofloxacin and piperacillin/tazobactam). This category includes *E. coli* (2 organisms) and *K. pneumoniae* (2 organisms) tested against ciprofloxacin, as well as *E. coli* (4 organisms) and *K. pneumoniae* (3 organisms) tested against piperacillin/tazobactam. Consequently, these organisms (9 classified as R and 2 as S, respectively, using the reference AST assay) were excluded from the calculation of CA and associated errors. These parameters were also not calculated for the levofloxacin/*P. aeruginosa* pair, as the highest concentration of antimicrobial in the Vitek Reveal AST assay was lower than the EUCAST R breakpoint.

**Table 2 antibiotics-13-01058-t002:** Evaluation of the Vitek Reveal AST assay results obtained with clinical positive blood culture samples for Gram-negative bacterial organisms.

	No. of Bacterial Organisms Categorized According to EUCAST Breakpoints v14.0 as:			Parameters Evaluated According to:					
				Both ISO 20776–2:2021 and ISO 20776–2:2007	ISO 20776–2:2021 Only	ISO 20776–2:2007 Only			
	R	I	S	EA (%; *n*/N)	Bias (%; No. of On-Scale Organisms)	CA (%; *n*/N)	ME (%; *n*/N)	VME (%; *n*/N)	mE (%; *n*/N)
*Enterobacterales* organisms (*n* = 43) tested against:									
Amikacin	6	–	37	97.7; 42/43	–	95.4; 41/43	5.4; 2/37	0.0; 0/6	–
Amoxicillin/clavulanic acid	15	–	27	95.2; 40/42	–	100; 42/42	0.0; 0/27	0.0; 0/15	–
Ampicillin	12	–	12	100; 24/24	–	100; 24/24	0.0; 0/12	0.0; 0/12	–
Aztreonam	14	0	25	100; 39/39	–	100; 39/39	0.0; 0/25	0.0; 0/14	0.0; 0/39
Cefepime	13	0	30	93.0; 40/43	–	100; 43/43	0.0; 0/30	0.0; 0/13	0.0; 0/43
Cefotaxime	14	0	29	100; 43/43	–	100; 43/43	0.0; 0/29	0.0; 0/14	0.0; 0/43
Ceftazidime	14	2	27	95.4; 41/43	−3.7; 11	95.4; 41/43	0.0; 0/27	0.0; 0/14	4.7; 2/43
Ceftazidime/avibactam	1	–	42	95.4; 41/43	–	100; 43/43	0.0; 0/42	0.0; 0/1	–
Ceftolozane/tazobactam	7	–	32	100; 39/39	–	100; 39/39	0.0; 0/32	0.0; 0/7	–
Ciprofloxacin	16	0	27	100; 43/43	–	100; 40/40	0.0; 0/26	0.0; 0/14	0.0; 0/40
Ertapenem	6	–	37	97.7; 42/43	–	100; 43/43	0.0; 0/37	0.0; 0/6	–
Gentamicin	10	–	33	100; 43/43	–	100; 43/43	0.0; 0/33	0.0; 0/10	–
Imipenem	5	0	34	100; 39/39	–	97.4; 38/39	0.0; 0/34	0.0; 0/5	2.6; 1/39
Levofloxacin	14	0	29	100; 43/43	–	97.7; 42/43	0.0; 0/29	0.0; 0/14	2.3, 1/43
Meropenem	5	0	38	97.7; 42/43	–	100; 43/43	0.0; 0/38	0.0; 0/5	0.0; 0/43
Meropenem/vaborbactam	1	–	42	100; 43/43	–	100; 43/43	0.0; 0/42	0.0; 0/1	–
Piperacillin	24	–	19	100; 43/43	–	100; 43/43	0.0; 0/19	0.0; 0/24	–
Piperacillin/tazobactam	11	–	32	100; 43/43	–	100; 43/43	0.0; 0/32	0.0; 0/11	–
Tigecycline	0	–	20	100; 20/20	–	100; 20/20	0.0; 0/20	0.0; 0/0	–
Tobramycin	11	–	32	100; 43/43	–	97.7; 42/43	0.0; 0/32	9.1; 1/11	–
Trimethoprim/sulfamethoxazole	14	0	29	100; 43/43	–	100; 43/43	0.0; 0/29	0.0; 0/14	0.0; 0/43
Total antibiotics	213	2	633	98.7; 837/848	−7.4; 52	99.2; 838/845	0.3; 2/632	0.5; 1/211	1.1; 4/376
*Pseudomonas aeruginosa* organisms (*n* = 6) tested against:									
Amikacin	0	–	6	100; 6/6	–	100; 6/6	0.0; 0/6	0.0; 0/0	–
Aztreonam	0	6	–	100; 6/6	–	100; 6/6	0.0; 0/6	0.0; 0/0	–
Cefepime	1	5	–	100; 6/6	–	100; 6/6	0.0; 0/5	0.0; 0/1	–
Ceftazidime	0	6	–	100; 6/6	–	100; 6/6	0.0; 0/6	0.0; 0/0	–
Ceftazidime/avibactam	0	–	6	100; 6/6	–	100; 6/6	0.0; 0/6	0.0; 0/0	–
Ceftolozane/tazobactam	0	–	6	100; 6/6	–	100; 6/6	0.0; 0/6	0.0; 0/0	–
Ciprofloxacin	1	5	–	100; 6/6	–	100; 6/6	0.0; 0/5	0.0; 0/1	–
Imipenem	0	6	–	100; 6/6	–	100; 6/6	0.0; 0/6	0.0; 0/0	–
Levofloxacin	1	5	–	100; 6/6	–	–	–	–	–
Meropenem	0	0	6	100; 6/6	–	100; 6/6	0.0; 0/6	0.0; 0/0	0.0; 0/6
Meropenem/vaborbactam	0	–	6	100; 6/6	–	100; 6/6	0.0; 0/6	0.0; 0/0	–
Piperacillin	0	6	–	100; 6/6	–	100; 6/6	0.0; 0/6	0.0; 0/0	–
Piperacillin/tazobactam	0	6	–	100; 6/6	–	100; 6/6	0.0; 0/6	0.0; 0/0	–
Tobramycin	0	–	6	100; 6/6	–	100; 6/6	0.0; 0/6	0.0; 0/0	–
Total antibiotics	3	45	36	100; 84/84	−36.6; 34	100; 78/78	0.0; 0/76	0.0; 0/2	0.0; 0/6
*Acinetobacter baumannii* organisms (*n* = 1) tested against:									
Amikacin	1	–	0	100; 1/1	–	100; 1/1	0.0; 0/0	0.0; 0/1	–
Ciprofloxacin	1	0	–	100; 1/1	–	100; 1/1	–	0.0; 0/1	–
Gentamicin	1	–	0	100; 1/1	–	100; 1/1	0.0; 0/0	0.0; 0/1	–
Imipenem	1	0	0	100; 1/1	–	100; 1/1	0.0; 0/0	0.0; 0/1	0.0; 0/1
Levofloxacin	1	0	0	100; 1/1	–	100; 1/1	0.0; 0/0	0.0; 0/1	0.0; 0/1
Meropenem	1	0	0	100; 1/1	–	100; 1/1	0.0; 0/0	0.0; 0/1	0.0; 0/1
Tobramycin	1	–	0	100; 1/1	–	100; 1/1	0.0; 0/0	0.0; 0/1	–
Trimethoprim/sulfamethoxazole	1	0	0	100; 1/1	–	100; 1/1	0.0; 0/0	0.0; 0/1	0.0; 0/1
Total antibiotics	8	0	0	100; 8/8	–	100; 8/8	0.0; 0/0	0.0; 0/8	0.0; 0/4
All organisms tested (*n* = 50)	224	47	669	98.8; 929/940	−11.0; 86	99.2; 924/931	0.3; 2/708	0.4; 1/221	1.0; 4/386

AST, antimicrobial susceptibility testing; CA, categorical agreement; EA, essential agreement; EUCAST, European Committee on Antimicrobial Susceptibility Testing; I, susceptible, increased exposure; ISO, International Organization for Standardization; ME, major error; mE, minor error; R, resistant; S, susceptible, standard dosing regimen; VME, very major error. Note: The symbol “–” indicates not applicable. Across the tested organism groups, the number of evaluated antimicrobials varies by bacterial species, as species known to exhibit intrinsic resistance to the antimicrobials included in the Vitek Reveal AST assay, or for which EUCAST breakpoints are unavailable, were excluded from the analysis. The EUCAST-defined category “area of technical uncertainty” only applies to two of the antibiotics listed here (ciprofloxacin and piperacillin/tazobactam). This category includes *E. coli* (1 organism) and *K. pneumoniae* (2 organisms) tested against ciprofloxacin, while none of the organisms tested against piperacillin/tazobactam were included in this category. Consequently, these organisms (2 classified as R and 1 as S, respectively, using the reference AST assay) were excluded from the calculation of CA and associated errors. These parameters were also not calculated for the levofloxacin/*P. aeruginosa* pair, as the highest concentration of antimicrobial in the Vitek Reveal AST assay was lower than the EUCAST R breakpoint.

## Data Availability

Data may be available upon reasonable request.

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
