# Peer review of "Verification of the Vitek Reveal System for Direct Antimicrobial Susceptibility Testing in Gram-Negative Positive Blood Cultures"

_antibiotics, 2024, doi:10.3390/antibiotics13111058_

Round 1
Reviewer 1 Report
Comments and Suggestions for Authors
The authors presented studies of the accuracy of the Vitek Reveal, the antimicrobial susceptibility testing assay. The Vitek Reveal AST test was validated according to ISO 20776-2:2021 and ISO 20776-2:2007 criteria. 100 simulated and clinical blood cultures containing Gram-negative bacteria were tested. Simulated GN-PBC was obtained from a hospital collection of isolates selected to represent different antimicrobial resistance profiles. The results showed that the Vitek Reveal AST assay provides rapid and reliable results. The results are important for microbiological diagnostics and antimicrobial susceptibility testing.
I would like to ask the authors to make the following corrections:
1. In Methods, the number of samples, the acceptance rates of samples for testing, and the rejection rates should be provided.
2. The manufacturers of reagents, media, etc. should be provided.
3. At the beginning, I propose adding a list of abbreviations used in the paper.
Author Response
Comments 1: The authors presented studies of the accuracy of the Vitek Reveal, the antimicrobial susceptibility testing assay. The Vitek Reveal AST test was validated according to ISO 20776-2:2021 and ISO 20776-2:2007 criteria. 100 simulated and clinical blood cultures containing Gram-negative bacteria were tested. Simulated GN-PBC was obtained from a hospital collection of isolates selected to represent different antimicrobial resistance profiles. The results showed that the Vitek Reveal AST assay provides rapid and reliable results. The results are important for microbiological diagnostics and antimicrobial susceptibility testing.
Response 1: I would like to thank the reviewer for appreciating our study and for giving us the opportunity to modify the manuscript
Comments 2: I would like to ask the authors to make the following corrections: In Methods, the number of samples, the acceptance rates of samples for testing, and the rejection rates should be provided.
Response 2: I modified the manuscript by adding this relevant information. See Methods, pages 11 and 12 of the revised manuscript.
Comments 3: The manufacturers of reagents, media, etc. should be provided.
Response 3: I modified the manuscript by adding this relevant information. See Methods, pages 12 and 13 of the revised manuscript.
Comments 4: At the beginning, I propose adding a list of abbreviations used in the paper.
Response 4: Unfortunately, the Antibiotics guidelines do not allow a list of abbreviations at the beginning of the paper. However, I made sure that all abbreviations were spelled out at the first mention.
Reviewer 2 Report
Comments and Suggestions for Authors
The authors present a laboratory validation of a recent rapid AST testing by Biomerieux, one of the leaders of automatized apparatus worldwide. These after-market independent studies are relevant to ensure the reproducibility of results without industry bias. The experiments were well conducted, and they used the reference standard (broth microdilution) to compare their results. The methods were well described. The text is well-written and easy to understand.
Only few minor errors were noted:
Line 17: Abstract: CA abbreviation is not previously described in the abstract,
The header in the last three columns of table 1 and table 2 describes (% n/N) but the percentage is not defined for all the rows.
Results: Does the authors have a hypothesis of why cefepime showed lower EA?, looking at the supplementary material, this antibiotic powder was supplied by USP instead of Sigma, as most of the other antibiotics. If the authors believe this is relevant may be included in the discussion
Discussion: Authors may consider adding to the discussion previous in-house attempts to obtain AST directly from positive blood cultures once a gram-negative microorganism was detected using Vitek AST cards (e.g. Bruins M, Bloembergen P, Ruijs G, Wolfhagen M. Identification and susceptibility testing
of Enterobacteriaceae and Pseudomonas aeruginosa by direct inoculation from positive BACTEC blood culture bottles into Vitek 2. J Clin Microbiol 2004;42) It could be discussed in terms of reproducibility and EA for which the Vitek Reveal may be superior but is somehow more comparable to broth microdilution, which is subculture-based.
Author Response
Comments 1: The authors present a laboratory validation of a recent rapid AST testing by bioMérieux, one of the leaders of automatized apparatus worldwide. These after-market independent studies are relevant to ensure the reproducibility of results without industry bias. The experiments were well conducted, and they used the reference standard (broth microdilution) to compare their results. The methods were well described. The text is well-written and easy to understand.
Response 1: I would like to thank the reviewer for appreciating our study and for giving us the opportunity to modify the manuscript.
Comments 2: Line 17: Abstract: CA abbreviation is not previously described in the abstract.
Response 2: The abbreviation CA was spelled out at the first mention. See page 1 of the revised manuscript.
Comments 3: The header in the last three columns of table 1 and table 2 describes (% n/N) but the percentage is not defined for all the rows.
Response 3: I completed both tables with missing information. See pages 3 to 8 of the revised manuscript.
Comments 4: Do the authors have a hypothesis of why cefepime showed lower EA? looking at the supplementary material, this antibiotic powder was supplied by USP instead of Sigma, as most of the other antibiotics. If the authors believe this is relevant may be included in the discussion.
Response 4: I appreciated the hypothesis suggested by the reviewer and added a short sentence of the issue in the Limitations section of the manuscript. See page 11 of the revised manuscript.
Comments 5: Authors may consider adding to the discussion previous in-house attempts to obtain AST directly from positive blood cultures once a gram-negative microorganism was detected using Vitek AST cards (e.g. Bruins M, Bloembergen P, Ruijs G, Wolfhagen M. Identification and susceptibility testing of Enterobacteriaceae and Pseudomonas aeruginosa by direct inoculation from positive BACTEC blood culture bottles into Vitek 2. J Clin Microbiol 2004;42) It could be discussed in terms of reproducibility and EA for which the Vitek Reveal may be superior but is somehow more comparable to broth microdilution, which is subculture-based.
Response 5: I added a few sentences to discuss our current results in the context of previous research performed by Bruins et al. and us (De Angelis et al.; https://doi.org/10.1093/jac/dky532) on direct testing of positive blood cultures using the Vitek AST assay. See page 10 of the revised manuscript.